# Blood CD8+ Naïve T-Cells Identify MS Patients with High Probability of Optimal Cellular Response to SARS-CoV-2 Vaccine

**DOI:** 10.3390/vaccines11091399

**Published:** 2023-08-22

**Authors:** Alexander Rodero-Romero, Susana Sainz de la Maza, José Ignacio Fernández-Velasco, Enric Monreal, Paulette Esperanza Walo-Delgado, Juan Luis Chico-García, Noelia Villarrubia, Fernando Rodríguez-Jorge, Rafael Rodríguez-Ramos, Jaime Masjuan, Lucienne Costa-Frossard, Luisa María Villar

**Affiliations:** 1Department of Immunology, Hospital Universitario Ramón y Cajal, Instituto Ramón y Cajal de Investigación Sanitaria (IRYCIS), Red Española de Esclerosis Múltiple (REEM), Red de Enfermedades Inflamatorias (REI), Universidad de Alcalá, 28034 Madrid, Spain; alexander.rodero@salud.madrid.org (A.R.-R.); jfvelasco@salud.madrid.org (J.I.F.-V.); pauletteesperanz.walo@salud.madrid.org (P.E.W.-D.); noelia.villarrubia@salud.madrid.org (N.V.); rafael.rodriguez.ramos@salud.madrid.org (R.R.-R.); 2Department of Neurology, Hospital Universitario Ramón y Cajal, Instituto Ramón y Cajal de Investigación Sanitaria (IRYCIS), Red Española de Esclerosis Múltiple (REEM), Universidad de Alcalá, 28034 Madrid, Spainenrique.monreal@salud.madrid.org (E.M.); juanluis.chico@salud.madrid.org (J.L.C.-G.); frjorge@salud.madrid.org (F.R.-J.); jaime.masjuan@salud.madrid.org (J.M.); lucienne.costa@salud.madrid.org (L.C.-F.)

**Keywords:** cellular immune response, SARS-CoV-2 vaccination, multiple sclerosis, disease-modifying therapies

## Abstract

This single-center study included 68 multiple sclerosis (MS) patients who received the severe acute respiratory syndrome coronavirus 2 (SARS-CoV-2) vaccination from one of several approved vaccine preparations in Spain. Blood samples were collected one to three months after the second dose of the vaccine had been administered. Cellular immune responses to the vaccine were assessed using QuantiFERON analysis, and peripheral blood mononuclear cell subsets were assayed using flow cytometry. Response associated with higher percentages of total lymphocytes, naïve CD4+ T-cells (*p* = 0.028), CD8+ T-cells (*p* = 0.013), and, mostly, naïve CD8+ T-cells (*p* = 0.0003). These results were confirmed by analyzing absolute numbers (*p* = 0.019; *p* = 0.002, and *p* = 0.0003, respectively). Naïve CD8 T-cell numbers higher than 17 cells/μL were closely associated with an optimal cellular response to SARS-CoV-2 vaccination (odds ratio: 24.0, confidence interval: 4.8–460.3; *p* = 0.0001). This finding clearly shows that independent of the treatment received, higher numbers of naïve CD8+ T-cells yield a strong cellular response to SARS-CoV-2 vaccines in MS patients. If this finding is validated with other viruses/vaccines, it could provide a good tool for identifying MS patients undergoing treatment who will develop strong cellular responses to anti-virus vaccines.

## 1. Introduction

Multiple sclerosis (MS) is the most frequent inflammatory disease of the central nervous system. Although its pathophysiology is still unknown, there are different disease-modifying therapies (DMTs) that change the course of MS by suppressing or modulating immune function that are available. DMTs change the natural history of the disease and contribute to improving patient outcomes [1]. Due to the mechanism of action of these therapies, it is important to evaluate how the immune system might respond to vaccines. Pre-vaccination may postpone treatment initiation, which may result in a worse disease outcome, particularly in active patients [2,3]. By contrast, vaccination of patients under treatment may result in a lack of effectiveness for the vaccine and thus, a higher risk of infections [3,4,5].

When it comes to severe acute respiratory syndrome coronavirus 2 (SARS-CoV-2) vaccinations, current guidelines recommend that all patients with MS, regardless of their DMT treatment status, receive the SARS-CoV-2 vaccine as soon as it is available to them. The benefits of SARS-CoV-2 vaccination outweigh the potential risks, and the vaccine is considered safe and effective for patients with MS [6,7,8]. However, some cases, such as patients of older age or those treated with fingolimod or anti-CD20 treatments, generally showed a suboptimal humoral immune response to the vaccine [9,10]. These suboptimal responses have resulted in an opportunity to explore the cellular immune cell profiles associated with responses to the SARS-CoV-2 vaccine in MS patients.

Although most studies analyze the effectiveness of vaccines through the production of antibodies [7,8], vaccines involve multiple lines of defense, mainly those mediated by cellular immune responses. These responses seem to be especially useful for avoiding a severe infection [9,10,11], as demonstrated in patients treated with anti-CD20 antibodies [12]. During the severe acute respiratory syndrome caused by the SARS-CoV-2 pandemic, the implementation of methods for measuring cellular immune responses allowed us to monitor these responses and try to elucidate their protective role in patients receiving different DMTs [7]. More patients showed protective cellular responses than humoral ones. This process was clearly demonstrated for patients who had been treated with anti-CD20 antibodies, who presented strong and long-lasting protective cellular responses in most cases [11]. Yet, it was found that a number of MS patients did not mount effective cellular responses to SARS-CoV-2 [13].

Identifying patients who have a high probability of mounting optimal cellular responses is of the utmost clinical importance. Our aim was to investigate whether any immune cell subsets could be used as biomarkers to identify potential responders.

## 2. Materials and Methods

### 2.1. Study Design

We performed a single-center cross-sectional study at the Ramón y Cajal University Hospital in Madrid. The study was approved by the Ethics Committee, and all participants signed an informed consent.

### 2.2. Patient Characteristics

Sixty-eight patients were included in this study. All of them received a complete vaccination cycle with one of the several SARS-CoV-2 vaccines approved in Spain. Blood samples were collected between one and three months after administration of the last dose of vaccine (second for vaccines mRNA-1273 (Moderna, Boston, MA, USA), BNT162b2 (Pfizer/BioNTech, New York, NY, USA), or ChAdOx1nCoV-19 (AZD1222, AstraZeneca, Cambridge, UK) or one for the only patient receiving vaccine Ad26.COV2-S (JNJ78436735, Johnson & Johnson, Old Brunswick, NJ, USA). The gap between the two doses of the vaccine was three weeks in the case of Pfizer/BioNTech vaccine, four weeks in the case of Moderna vaccine, and 10 weeks in the case of AstraZeneca vaccine.

Clinical data are shown in Table 1. Median age was 41.05 years, 42 (61.8%) patients were women, and 64 (94.1%) were receiving some form of DMT at the time of vaccination. DMTs were divided into four classes of treatment: (1) none, if they were not currently receiving any treatment at sampling; (2) pulsed treatment, including cladribine and alemtuzumab; (3) anti-CD20 antibodies, including ocrelizumab and rituximab; and (4) continuous treatments, including fingolimod, natalizumab, dimethylfumarate, glatiramer acetate, and/or teriflunomide.

Sixty-two patients (91.1%) were vaccinated with both doses of an mRNA vaccine (mRNA-1273 or BNT162b2), five (7.4%) were vaccinated with both doses of ChAdOx1nCoV-19, and one (1.5%) was vaccinated with a dose of Ad26.COV2-S.

We monitored the years elapsed since DMT initiation to the first vaccine dose and since the last drug administration for pulsed immune therapies and anti-CD20 therapies (Table 2).

### 2.3. Sample Collection

Peripheral blood mononuclear cells (PBMCs) were isolated from 20 mL of heparinized blood using Ficoll density gradient centrifugation (Abbott Laboratories, Chicago, IL, USA) and cryopreserved until analysis. Total lymphocyte and monocyte counts were determined in fresh ethylenediaminetetraacetic acid (EDTA)-treated blood using a Coulter counter.

### 2.4. Cell Cultures and Interferon-γ Quantification

PBMCs were thawed and resuspended in RPMI medium supplemented with 10% fetal calf serum and 1 mM glutamine (Merck, Darmstadt, Germany), which was considered complete medium, at a concentration of 5.0 × 10^6^ cells/mL. Aliquots of 200 µL were cultured in 96-well flat-bottom plates (Corning Incorporated, New York, NY, USA). Three different culture conditions were employed for each patient: (1) PBMCs incubated in complete medium were used as a negative control, (2) 10 μL (10 μg/mL) of OKT3 anti-CD3 antibody (BD Biosciences, New York, NY, USA) was added to PBMCs as a positive control, and (3) 4 µL (50 µg/mL) of spike protein (S) peptide from SARS-CoV-2 (PepTivator, SARS-CoV-2 Prot S, Miltenyi Biotec, Bergisch Gladbach, Germany) to identify interferon gamma (IFN-γ) production by T-lymphocytes in response to stimulation with SARS-CoV-2 spike peptides. After incubating at 37 °C, 5% CO_2_, and 95% humidity for 30 min, 6 µL (25 µg/mL) of anti-CD28/CD49d co-stimulator antibody (BD Biosciences) was added to each well, and cells were incubated for 24 h at 37 °C, 5% CO_2_, and 95% humidity. After incubation, plates were centrifuged, and supernatants were collected and stored at −80 °C until analyzed.

We used single-molecule array (SIMOA) IFN-γ advantage kit technology (Quanterix, Billerica, MA, USA) to quantify IFN-γ levels in supernatants using an SR-X instrument (Quanterix, Billerica, MA, USA). Positive responses were defined as previously published (13). Briefly, IFN-γ levels ≥ 80 pg/mL upon stimulation with spike peptides were considered positive.

### 2.5. Monoclonal Antibodies

The monoclonal antibodies used in this study included CD3 PerCP, CD5 APC-R700, CD8-FITC, CD11c PE-Cy5, CD14 APC-H7, CD19 PE-Cy5, CD25 PE-Cy7, CD24 FITC, CD27 PE-Cy7, CD28 PE-Cy5, CD38 APC-H7, CD45 V500, CD45RO APC, CD56 APC, CD57 FITC, CD80 PE, CD86 APC, CD123 APC, CD127 BV421, CCR7 PE, TIGIT BV421, HLA DR V450, TIM3 PE, CXCR5 APC-R700, PD1 BV421, and PDL1 PE-Cy7 (BD Biosciences).

### 2.6. Labeling of Surface Antigens

After a period of one to six months, cryopreserved PBMCs were thawed, and viability was evaluated in a Neubauer Chamber by using Trypan Blue dye exclusion test (Merck). 200.000 viable cells per tube were labeled with appropriate amounts of fluorescently-labeled monoclonal antibodies for 30 min at 4 °C in the dark. Cells were washed twice with phosphate-buffered saline (PBS) and analyzed by flow cytometry as described below.

### 2.7. Flow Cytometry

A minimum number of 10 × 10⁴ events were analyzed. Gating strategies for identifying the different subsets of T- and B-cells, monocytes, natural killers (NKs), and dendritic cells (DC) are shown in Appendix A.

### 2.8. Flow Cytometry Analyses

We recorded for every leukocyte subset total cell counts per mL of blood, calculated by measuring total lymphocyte and monocyte numbers by a Coulter Counter, and the percentages of every subset over total mononuclear cells.

### 2.9. Serum Anti-Spike Antibodies

Serum IgG antibodies against the spike protein of SARS-CoV-2 (S1 subunit) were studied by a chemiluminescence immunoassay of micro-particles (ALINITY system, Abbott Laboratories, Chicago, IL, USA) as previously described. Levels of binding antibody units per milliliter (BAU/mL) higher than 260/mL were considered protective.

### 2.10. Statistical Analysis

Statistical analyses were performed using GraphPad Prism 9.0 software (GraphPad Prism Inc., San Diego, CA, USA). A Mann–Whitney U-test was used to compare continuous variables and Fisher’s exact tests to analyze categorical ones. Cutoff values were established by estimating the Youden index using receiver operating characteristic (ROC) curve analyses. *p* values < 0.05 were considered significant.

## 3. Results

### 3.1. Patients

We classified patients according to their cellular response as responders (61 patients, 59% females) and non-responders (seven patients, 86% females) based on the T-cell production of IFN-γ against spike peptides. Clinical and demographic data of responder and non-responder groups are shown in Table 3. We did not find significant differences in disease duration between both groups. However, age at vaccination was lower in responders when compared to non-responders (*p* = 0.049). We also observed significant differences in the DMTs received by the patients after vaccination administration with a lower proportion of patients treated with fingolimod in the responder group (*p* = 0.0013).

### 3.2. Percentages of the Different Peripheral Blood Immune Cell Subsets

We examined the different PBMC subsets in both groups of patients. Results are shown in Table 4. Responders had a higher percentage of lymphocytes (*p* = 0.0043). We investigated the different T- and B-cell subsets to determine the subsets associated with this increase.

#### 3.2.1. T-Cells

No differences were found in total CD4+ T-cells, but responders showed higher percentages of naïve CD4+ T-cells (*p* = 0.028, Figure 1) and a trend toward a higher number of CD4+ effector memory cells (*p* = 0.056). Responders also showed higher values of total T CD8+ cells (*p* = 0.013, Figure 1) due to a marked increase in naïve CD8+ cells (*p* = 0.0003, Figure 1). We did not find other significant changes in other CD8+ T-cell subsets.

Additionally, we did not find significant differences in immunosenescent CD4 and CD8 T-cells or in those expressing T-cell immunoglobulin and mucin-containing protein 3 (TIM3) or T-cell receptor with Ig and the immunoreceptor tyrosine-based inhibitory motif (TIGIT), as shown in Table 4.

#### 3.2.2. B-Cells

No significant differences were observed in total B-cells or in the different B-cell subsets between patient groups.

#### 3.2.3. Innate Immune Cells

Responders showed a lower proportion of monocytes (*p* = 0.0036), myeloid cells (*p* = 0.0075), and plasmacytoid dendritic cells (*p* = 0.013).

### 3.3. Absolute Cell Counts

We next assessed whether the differences between responders and non-responders were total or relative. We evaluated the total cell numbers in all subsets giving significant differences in the study of the percentages (Table 4). We still observed differences in the total absolute number of lymphocytes (*p* = 0.0088), naïve CD4+ T-cells (*p* = 0.0189), total CD8+ T-cells (*p* = 0.0019), and naïve CD8+ T-cells (*p* = 0.0003). However, the decrease in monocytes and dendritic cells was only relative, probably due to the higher lymphocyte counts.

### 3.4. Cutoff Values

We next explored whether any of the subpopulations could be considered a useful tool for identifying patients with a higher possibility of having a good cellular immune response to the virus. We selected naïve CD8 T-cells because the clearest differences between responders and non-responders were found in this T-cell subset. Using an ROC curve analysis, we established a cutoff value of 1.91% for percentages (area under the curve [AUC] = 0.89, *p* = 0.0009; sensitivity = 76.7, specificity = 85.7) and of 17 cells/μL for the total numbers (AUC = 0.94, *p* = 0.0002; sensitivity = 88.7, specificity = 100). We stratified the patients using the last value and analyzed the results using a Fisher exact test. Results are shown in Figure 2. A total of 87% of responders (46 out of 53) had values higher than the cutoff value, while only 14% of non-responders (1 out of 7) reached these values (odds ratio = 24.0, confidence interval: 4.8–460.3; *p* = 0.0001).

To further explore this, we studied the response to the third dose of anti SARS-CoV-2 vaccines in three patients showing a CD8+ naïve T-cell number higher than the cutoff value and a good cellular response to the second dose, as well as in two patients presenting the opposite situation. The three patients with high naïve CD8+ T-cell numbers again showed an optimal cellular immune response, with a median IFN-γ value of 3271.8 pg/mL. Those patients with low numbers of these types of cells showed a poor response, with a median IFN-γ value of 4.2 pg/mL. This finding strongly suggests that naïve CD8+ T-cell numbers may be a good predictor of cellular responses to these vaccines.

### 3.5. Correlation of T-cells and IFN-γ Production in Response to Spike Antigen

The production of IFN-gamma by T-cells was assessed by culturing PBMC from every patient as described previously [13]. We next explored the association between values of IFN gamma in culture supernatants and native CD8 cells. We found a correlation between the percentages (r = 0.42, *p* = 0.0004, and the numbers r = 0.41, *p* = 0.0006) of naïve CD8 T-cells and IFN-Y levels, thus suggesting there is a relationship between these cells and IFN- γ production.

### 3.6. Humoral Responses

This was monitored by evaluating titers of anti-spike IgG antibodies. A total of 21 patients (32.3% of the total group) did not reach protective levels. Of these, 5 were treated with fingolimod (71.4% of the total fingolimod-treated group); 15 received anti-CD20 antibodies (71.4% of the anti-CD20-treated group), and the remaining 1 was given cladribine (9.1% of the total cladribine-treated group).

## 4. Discussion

The effects of DMTs on the immune response to different vaccines have been widely studied in MS. These treatments, which have immunomodulatory/immunosuppressive activities, can increase the risk of new infections [14,15]. Vaccination plays a crucial role in preventing infections in patients with DMTs, and because of the possibility of infections, it is recommended to vaccinate patients whenever possible before initiating any treatment [15,16]. Evidence suggests that inactivated vaccines do not increase the risk of relapses and are safe for MS patients even during treatment [16]. However, the immune response to vaccines may be diminished, particularly in individuals receiving immunosuppressive DMTs [15,16]. To assess the patient’s immune status and evaluate the potential risks and benefits of vaccination, a serological study and the vaccination against different viruses is individually recommended prior to treatment initiation; however, this process may result in a delay in treatment initiation and an increase in the risk of disease worsening in the interim [14,15,16].

Typically, the evaluation of vaccination risks and patient responses is based on examining humoral responses [15]. However, cellular responses also play a crucial role in preventing infection, as observed with SARS-CoV-2 vaccines [17,18,19]. In MS patients treated with anti-CD20 antibodies, who generally show low levels of anti-spike antibodies, there is no correlation between the antibody titers and IFN-γ amounts, as they showed a good cellular response [13,19,20]. Most MS patients show a good cellular response to SARS-CoV-2 vaccines but different issues, such as age [21] or type of DMT [22], can also alter the cellular immune response to the vaccines. Thus, monitoring the cellular response would be important to explore the immune status against different pathogens in MS and the response to vaccines. However, this seems difficult in the cases of other microorganisms, different from SARS-CoV-2, where there are no specific QuantiFERON kits and for which we do not even know the immunodominant peptides to assay IFN-γ production by T-cells. A plausible alternative should involve investigating whether a cellular phenotype is associated with optimal immune responses.

This study aimed to evaluate the immune cell profile associated with the cellular response to SARS-CoV-2 vaccines in MS patients treated with DMTs. We chose SARS-CoV-2 as proof of concept due to the availability of commercial kits to assess cellular immune responses. Additionally, a considerable number of MS patients treated with different DMTs have been vaccinated against SARS-CoV-2, providing an opportunity to study the different immune cell subsets associated with the cellular response to the vaccine.

As previously described [13,19,20,23], most MS patients show a robust cellular response to SARS-CoV-2 vaccines, with fingolimod [24] and older age [25] presenting as the main causes of suboptimal response.

Optimal responses were found to be associated with higher percentages and numbers of total lymphocytes, CD8+ T-cells, naïve CD4+ T-cells, and particularly of naïve CD8+ T-cells. Numbers of these cells above 17 cells/μL were found to be associated with a higher probability of having a good response to the vaccine. This finding fits with the causes of non-response since individuals with MS may show early immunosenescence, which is associated with a decline in naïve CD4+ and CD8+ T-cell numbers [22]. Similarly, fingolimod-treated patients retain naïve CD4+ and CD8+ T-cells in the lymph nodes [24,26].

These findings strongly suggest that naïve CD8+ T-cell numbers can predict the cellular response to vaccination in MS patients. Results should be validated in a larger multicenter cohort. The main limitation of the study is the lack of patients vaccinated against other viruses. The lack of reliable tests for performing analyses of the cellular response to other viruses was a limiting factor. Future studies will demonstrate the value of this cell subset to predict cell response to other viruses in MS.

## 5. Conclusions

Despite its limitations, our study provides valuable insights into the immune cell subsets as useful indicators for predicting responses to vaccinations in MS patients who are undergoing treatment with different DMTs. These findings can contribute to the development of optimal immunization strategies for MS patients treated with these drugs.

## Figures and Tables

**Figure 1 vaccines-11-01399-f001:**
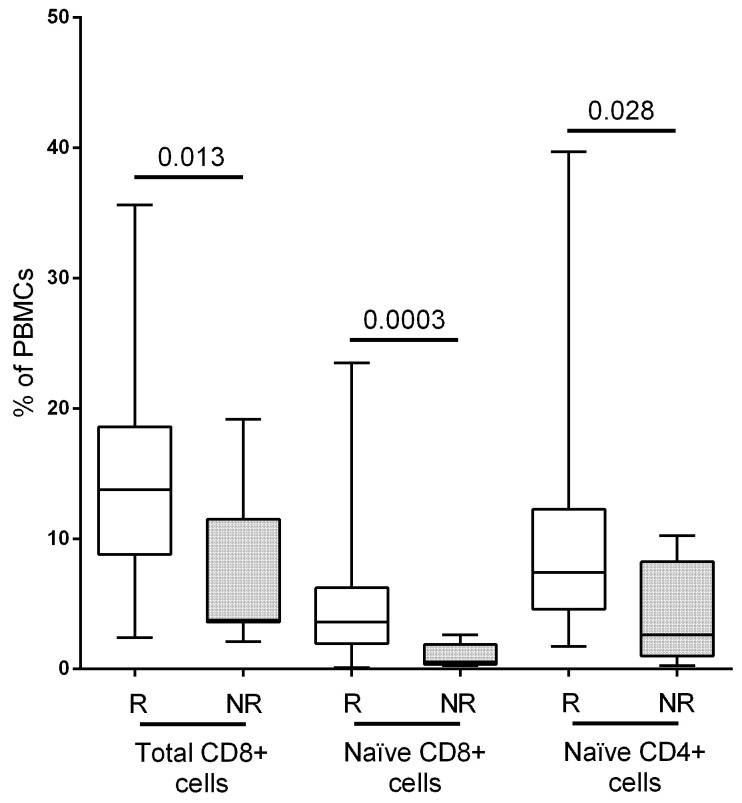
Percentage and *p*-value of total and naïve CD8+ T-cells and naïve CD4+ T-cells. The percentage of the subpopulations has been shown on CD45+ cells. R: responder group; NR: non-responder group.

**Figure 2 vaccines-11-01399-f002:**
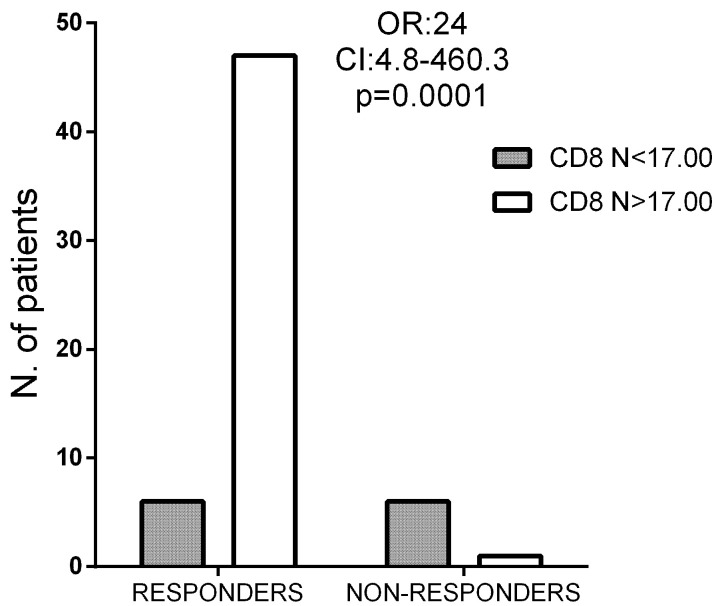
CD8+ Naïve T-cell Cutoff in cells/μL using absolute number of the subpopulation. In responders group, 47 patients had more cells/mL than the established cutoff, while 6 had fewer. Six patients in the non-responders did not go over the cutoff, while one patient did. ROC curve and Fisher exact test analyses were used in both graphs. AUC > 0.85 and *p* < 0.05 were considered significant. OR: odds ratio; CI: confidence interval; R: responder group; NR: non-responder group.

**Table 1 vaccines-11-01399-t001:** Demographic and clinical characteristics.

Characteristics	Total Population (*n* = 68)
Age, median [IQR (years)]	41.1 [32.9–51.9]
Females, *n* (%)	42.0 (61.8)
Time since first MS Symptoms, median [IQR (years)]	9.5 [6.1–16.2]
MS phenotype, *n* (%)	
Relapsing-remitting	53 (77.9)
Secondary progressive	9 (13.2)
Primary progressive	6 (8.8)
DMT, *n* (%)	
None	4 (5.9)
Pulsed treatments	
Alemtuzumab	13 (19.1)
Cladribine	11 (16.2)
Anti-CD20 treatments	
Ocrelizumab	13 (19.1)
Rituximab	8 (11.8)
Continuous treatments	
Fingolimod	7 (10.3)
Natalizumab	5 (7.3)
Dimethylfumarate	3 (4.4)
Platform (GA, IFN beta, teriflunomide)	4 (5.9)
SARS-CoV-2 vaccine, *n* (%)	
BNT162b2 (Pfizer/BioNTech)	13 (19.1)
mRNA-1273 (Moderna)	49 (72.0)
AZD1222 (AstraZeneca)	5 (7.4)
JNJ78436735 (Johnson & Johnson)	1 (1.5)

IQR: interquartilic range; MS: multiple sclerosis; DMT: disease-modifying therapy; GA: glatiramer acetate; IFN beta: interferon beta 1-alpha.

**Table 2 vaccines-11-01399-t002:** Treatment characteristics at the first vaccine dose.

DMT	Treatment Duration, Years [Median, IQR]	Time Since Last Infusion, Years [Median, IQR]
Pulsed Treatments Alemtuzumab Cladribine	3.5 [2.4–4.0] 0.5 [0.3–1.8]	1.9 [1.4–2.7] 0.3 [0.2–0.6]
Anti-CD20 treatments Ocrelizumab Rituximab	1.8 [0.8–2.9] 1.8 [0.6–2.6]	0.3 [0.2–0.4] 0.5 [0.4–0.6]
Continuous Treatments Fingolimod Natalizumab Dimethylfumarate Platform (GA, IFN beta, TF)	6.4 [4.5–7.3] 2.2 [0.2–8.6] 4.9 [3.3–5.3] 6.4 [4.5–7.3]	NA NA NA NA

IQR: interquartilic range; DMT: disease-modifying therapy; GA: glatiramer acetate; IFN beta: interferon beta; TF: teriflunomide; NA: not applicable.

**Table 3 vaccines-11-01399-t003:** Clinical and demographic characteristics after dividing the patients according to the production of IFN-γ.

	Responders (*n* = 61)	Non-Responders (*n* = 7)	*p* Value
Age at vaccionation onset (years), median (IQR)	39.6 (32.3–51.4)	50.4 (40.4–60.0)	0.049
Sex (male/female)	25/36	1/6	NS
Disease duration (years), median (IQR)	12.2 (8.8–21.0)	9.2 (5.4–15.2)	NS
MS phenotype, *n* (%) Relapsing-remitting Secondary progressive Primary progressive	48 (78.7) 8 (13.1) 5 (8.2)	5 (71.4) 1 (14.3) 1 (14.3)	NS
DMT (*n*)			0.0013
Pulsed treatments			
Alemtuzumab	13	0	
Cladribine	10	1	
Anti-CD20 treatments			
Ocrelizumab	12	1	
Rituximab	7	1	
Continuous treatments			
Fingolimod	3	4	
Natalizumab	5	0	
Dimethylfumarate	3	0	
Platform (GA, IFN beta, TF)	4	0	
None	4	0	

IQR: interquartilic range; DMT: disease-modifying therapy; GA: glatiramer acetate; IFN beta: interferon beta; TF: teriflunomide; ns: not significant

**Table 4 vaccines-11-01399-t004:** Percentage and absolute cell numbers of different immune cell subsets.

Cell Subset	Responders (*n* = 67) (%, Median, IQR)	Non-Responders (*n* = 7) (%, Median, IQR)	*p* Value	Responders (*n* = 67) (AN/µL, Median, IQR)	Non-Responders (*n* = 7) (AN/µL, Median, IQR)	*p* Value
Monocytes	14.5 (9.6 18.1)	31.6 (15.7–41.2)	0.0036	453 (102–540)	1250 (525–1950)	0.023
Lymphocytes	85.9 (81.9–90.2)	68.3 (58.8–84.3)	0.0043	1680 (955–2145)	640 (500–1540)	0.008
CD4+ T-cells	40.2 (30.3–50.9)	10.1 (6.5–51.2)	0.138			
Naïve	7.4 (4.6–12.3)	2.6 (1–8.2)	0.028	124 (71.5–292.5)	28.3 (9.3–188.7)	0.019
CM	12.7 (5.5–26.7)	1.9 (1.2–40.2)	0.191			
EM	5.9 (3.7–7.8)	3.2 (1.2–4.8)	0.056			
TD	5.8 (3.3–8.7)	4.7 (2.4–11.2)	0.834			
Treg	0.6 (0.3–2.5)	1.4 (0.4–2.6)	0.465			
TIGIT+	0.6 (0.4–0.9)	0.3 (0.1–1.1)	0.237			
Senescent	1.2 (0.7–1.9)	1.1 (0.2–2.1)	0.369			
TIM3+	0.7 (0.4–1.3)	1.0 (0.7–1.4)	0.503			
CD8+ T-cells	13.8 (8.8–18.6)	3.8 (3.6–11.5)	0.0135	231 (158–449)	77.5 (34.6–132.9)	0.0019
Naïve	3.6 (1.9–6.2)	0.6 (0.4–1.9)	0.0003	73.6 (32.5–133.1)	7.7 (3.8–15.9)	0.0003
CM	0.4 (0.2–0.7)	0.3 (0.1–0.5)	0.210			
EM	1.8 (0.8–3.3)	0.7 (0.4–2.5)	0.156			
TD	6 (4.1–8.8)	2 (1.2–7.9)	0.075			
TIGIT+	0.3 (0.2–0.7)	0.2 (0.1–0.5)	0.325			
Senescent	1.9 (0.8–4)	1.9 (0.3–5.8)	0.814			
TIM3+	2.1 (0.1–3.6)	1.9 (1.1–4.7)	0.526			
CD19+ cells	3.9 (0.1–9.6)	1.7 (1.4–5.1)	0.296			
Naïve	3.3 (0.1–8.3)	1.3 (0.9–4.2)	0.251			
Memory	0.4 (0.1–1)	0.2 (0.1–0.3)	0.300			
PBs	0.01 (0–0.02)	0.01 (0.01–0.03)	0.390			
TR	0.08 (0.01–0.2)	0.03 (0.22–0.65)	0.150			
NK cells						
NKT cells	1.7 (0.7–2.9)	1.5 (0.8–8.7)	0.616			
Bright	1.1 (0.5–1.7)	0.8 (0.7–1.3)	0.558			
Dim	12 (8.1–19)	18.8 (10.9–25.2)	0.081			
DCs						
PC	0.16 (0.1–0.3)	0.9 (0.2–1)	0.0132	3.1 (1.4–5.2)	7.2 (4.1–8.8)	0.108
Myeloid	0.3 (0.1–1.7)	3.3 (0.2–4.3)	0.0075	5.5 (2.5–31.1)	4.9 (3.4–13.5)	0.701

IQR: interquartile range; AN: absolute cell numbers; CM: central memory; EM: effector memory; TD: terminally differentiated; TIGIT: T-cell immunoglobulin and ITIM domain; TIM3: T-cell immunoglobulin and mucin domain-containing protein 3; PBs: plasmablasts; TR: transitional; NK: natural killer; DCs: dendritic cells; PC: plasmacytoid.

## Data Availability

Original data will be available to any researcher in the field during three years by request to the corresponding author.

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
