# Peer review of "Blood CD8+ Naïve T-Cells Identify MS Patients with High Probability of Optimal Cellular Response to SARS-CoV-2 Vaccine"

_vaccines, 2023, doi:10.3390/vaccines11091399_

Round 1

Reviewer 1 Report

The work aimed to evaluate the response of the vaccine to SARS-CoV-2 in patients with MS.  The work is relevant because most countries use only  the Pfizer vaccine in this population. In the present study, the vaccines BNT162b2 (BioNTech), mRNA-1273 (Moderna), AZD1222 (AstraZeneca), JNJ78436735 (Johnson & Johnson) were administered. Another point was that the study used different types of treatment such as Alemtuzumab Cladribine . Anti-CD20 treatments Ocrelizumab, Rituximab . Continuous treatments Fingolimod ,Natalizumab Dimethyl fumarate platform (GA, IFN beta, TNF). In some cases such as elderly patients the humoral response did not reach the expected response . This allowed the study to evaluate the cellular immune cell profiles associated with SARS-CoV-2 vaccine responses in MS patients. The study is relevant due to different vaccines administered compared to different immunosuppressive treatments for disease control in different age groups. The study shows a profile of stimulation of the cellular immune response in patients where the humoral response is decreased due to the treatment used. It also shows the importance of using Sars-CoV-2 vaccines and others  in this extremely fragile population.

There are some technical terms described in the article that were most likely misspelled.

TN= TNF 

 Review the references. I identified  two that are together! 

Review the references cited in the article 

20. Liebers N, Speer C, Benning L, et al. Humoral and cellular responses after COVID-19 vaccination in anti-CD20-treated lym- 359 phoma patients. Blood. 2022;139(1):142-147. doi:10.1182/blood.202101344521.XX TürkoÄŸlu R, Baliç N, Kızılay T, et al. Fingolimod 360 impairs inactivated vaccine (CoronaVac)-induced antibody response to SARS-CoV-2 spike protein in persons with multiple 361 sclerosis. Mult Scler Relat Disord. 2022;58:103524. doi:10.1016/j.msard.2022.103524

Author Response

Comment 1: There are some technical terms described in the article that were most likely misspelled: TN= TNF.

Answer: The term “TN” in the manuscript indicated treatment with teriflunomide, a first line therapy used in relapsing remitting MS. We made sure the abbreviation is indicated along the manuscript.

Comment 2: Review the references. I identified two that are together.

  1. Liebers N, Speer C, Benning L, et al. Humoral and cellular responses after COVID-19 vaccination in anti-CD20-treated lymphoma patients. Blood. 2022;139(1):142-147. doi:10.1182/blood.202101344521.XX TürkoÄŸlu R, Baliç N, Kızılay T, et al. Fingolimod 360 impairs inactivated vaccine (CoronaVac)-induced antibody response to SARS-CoV-2 spike protein in persons with multiple 361 sclerosis. Mult Scler Relat Disord. 2022;58:103524. doi:10.1016/j.msard.2022.103524.

Answer: We acknowledge the reviewer his observation. The second reference (TürkoÄŸlu R et al, 2022) should have been eliminated in the final version of the manuscript. We deleted it from the revised version.

Comment 3: Review the references cited in the article.

Answer: As suggested by the Reviewer, we checked the references cited in the manuscript and found a mistake in a reference included in the Introduction Section: where it says: [35], it should say: [3-5]. We modified it in the revised manuscript.

Reviewer 2 Report

Thanks to authors that they conducted the nice study to address the cellular response after vaccination in MS patients. However, putting anti-SARS-CoV-2 spike antibody titer will be necessary to meet the complete spectrum of this study.   

Few concerns need to be addressed before final publication:

1.    In line #74-75 it is mentioned as “samples were collected between one and three months after administration of the second dose of the vaccine” Whereas in line #85 it is mentioned as “one (1.5%) was vaccinated with a dose of Ad26.COV2-S.”. Make the sentence clear whether it was one or two dose of Ad26.COV2-S vaccination.

2.    What was the gap between two doses of different vaccination?

3.    What was the gap between blood sample collection and 2nd dose of vaccination?

4.     It would be important to mentioned how many PBMCs were stained for the flowcytometry analysis? And how much blood was collected from patients for PBMCs separation.  Further expressing absolute count of cell as “cell counts per µl of blood” would not be an appropriate representation rather expressing values in %, as cells were analyzed in PBMCs rather in whole blood.  If whole blood is used Kindly mentioned it in methodology.

5.    Did authors check the viability of PBMCs after thawing? And also mention the duration how long it was cryopreserved?

6.    Was there any correlation between naïve CD8+T cell and IFN-ϒ LEVEL?

7.    Did authors measure the humoral response (anti-SARS-COv-2 spike antibody) after vaccination? Did all MS patients form antibody after vaccination? It would be also important to measure humoral response together with cellular response. Humoral response is ultimately the goal of vaccination.  This will further strengthen the study. 

8.    Was there any association between the gap of anti-CD20 therapy and the first dose of vaccination in terms of IFN-ϒ level compared to those who did not receive anti-CD20 antibody therapy or those who have a higher gap between anti-CD20 therapy and the first dose of vaccination?  

English is fine.

Author Response

  1. In line #74-75 it is mentioned as “samples were collected between one and three months after administration of the second dose of the vaccine” Whereas in line #85 it is mentioned as “one (1.5%) was vaccinated with a dose of Ad26.COV2-S.”. Make the sentence clear whether it was one or two dose of Ad26.COV2-S vaccination.

Answer: As suggested by the reviewer, we clarified if one or two dose of COVID-19 vaccines were administrated, by including the following sentence under Patient Characteristics section: “Blood samples were collected between one and three months after administration of the last vaccine  dose (second for vaccines mRNA-1273 (Moderna), BNT162b2 (Pfizer/BioNTech), or ChAdOx1nCoV-19 (AZD1222, AstraZeneca, or one for a patient receiving vaccine Ad26.COV2-S (JNJ78436735, Johnson & Johnson).

  1. What was the gap between two doses of different vaccination?

Answer to Comment 2: We included the following sentence under Patient Characteristics section: “The gap between the two doses of the vaccine was three weeks in case of Pfizer/BioNTech vaccine, four weeks in the case of Moderna vaccine and 10 weeks in case of AstraZeneca vaccine.”

  1. What was the gap between blood sample collection and 2nd dose of vaccination?

Answer to Comment 3: As mentioned in the Patient Characteristics section of the manuscript, blood samples were collected between one and three months after administration of the last dose of vaccination.

4.1. It would be important to mention, how many PBMCs were stained for the flow cytometry analysis?

Answer: To address reviewer query, we included the following sentence under Labeling of surface antigens section of the manuscript: “200.000 viable cells per tube were labeled for 30 min at 4 °C in the dark with appropriate amounts of monoclonal antibodies labeled with different fluorochromes.”

4.2. And how much blood was collected from patients for PBMCs separation? 

Answer: We included the following sentence under Sample Collection section of the manuscript: “Peripheral blood mononuclear cells (PBMCs) were isolated from 20 ml of heparinized blood using Ficoll density gradient centrifugation (Abbott Laboratories, Chicago, IL, USA) and cryopreserved until analysis.”

4.3. Further expressing absolute count of cell as “cell counts per µl of blood” would not be an appropriate representation rather expressing values in %, as cells were analyzed in PBMCs rather in whole blood.  If whole blood is used, kindly mentioned it in methodology.

Answer: As mentioned by the reviewer, we analyzed the percentages in PBMC. However, we also analyzed total cell counts in parallel from a second aliquot of 5 mL fresh EDTA-treated blood using a Coulter Counter at sample collection. This allowed us to calculate total cell counts for every cell subset. To explain this we included the following sentence under Flow cytometry analyses in methods section: “We recorded for every leukocyte subset total cell counts per mL of blood, calculated by measuring total lymphocyte and monocyte numbers by a Coulter Counter, and the percentages of every subset over total mononuclear cells”. 

5.1. Did authors check the viability of PBMCs after thawing?

Answer: To explain how we evaluated the viability of PBMCs after thawing, we included the following sentence under Labeling of surface antigens section of the manuscript: “PBMCs were thawed, and viability evaluated in a Neubauer Chamber by using Trypan Blue dye exclusion test (Merck).”

5.2    And also mention the duration how long it was cryopreserved?

Answer: We mentioned under Labeling of surface antigens section of the manuscript how long PBMCs were cryopreserved: “After a period of one to six months, cryopreserved PBMCs were thawed (…)”.

  1. Was there any correlation between naïve CD8+T cell and IFN-ϒ LEVEL?

Answer: We explored this and found a correlation between IFN-Y levels and the percentages and total numbers of naïve CD8 T cells. We added the following sentence in the results section to show this: “Production of IFN-gamma by T cells was assessed by culturing PBMC from every patient as as described previously [13]. We next explored the association of values of IFN gamma in culture supernatants and native CD8 cells. We found a correlation between the percentages (r=0.42, p=0.0004, and the numbers r=0.41, p=0.0006) of naïve CD8 T cells and IFN-Y levels, thus suggesting there is a relationship between these cells and IFN- γ production”.

  1. Did authors measure the humoral response (anti-SARS-COv-2 spike antibody) after vaccination? Did all MS patients form antibody after vaccination? It would be also important to measure humoral response together with cellular response. Humoral response is ultimately the goal of vaccination. This will further strengthen the study.

Answer: We also measured humoral response. As we previously showed in other cohort (Sainz de la Maza S. et al. Vaccines 2023, 11, 786, Ref 13 of this manuscript) most patients showed a good antibody response to SARS-CoV-2 vaccines, with the exception of those treated with fingolimod or anti CD20 antibodies. To fulfil reviewer requirements we showed antibody data by including the following paragraph under results: “Humoral response was monitorized by evaluating titers of anti-spike IgG antibodies. Twenty-one patients (32.3% of the total group) did not reach protective levels. Five were treated with fingolimod (71.4% of total fingolimod-treated group); 15 were receiving with anti CD20 antibodies (71.4% of the anti CD20-treated group), and the remaining one cladribine (9.1% of the total cladribine- treated group)".

We also added this paragraph under methods:

Serum Anti-Spike Antibodies

Serum IgG antibodies against the spike protein of SARS-CoV-2 (S1 subunit) were studied by a chemiluminescence immunoassay of micro-particles (ALINITY system, Abbott Laboratories, Chicago, IL, USA) as previously described.  Levels of binding antibody units per milliliter (BAU/mL) higher than 260/ml were considered as protective.

  1. Was there any association between the gap of anti-CD20 therapy and the first dose of vaccination in terms of IFN-ϒ level compared to those who did not receive anti-CD20 antibody therapy or those who have a higher gap between anti-CD20 therapy and the first dose of vaccination?

Answer:  Different articles have demonstrated that T and B cell responses to SARS-CoV-2 vaccines in MS patients treated with anti CD20 antibodies are divergent. Very few of them show good serological responses but most have good cellular ones. By contrast, patients treated with fingolimod show poor serological and cellular responses and untreated patients or those treated with other drugs, generally show good serological and cellular responses. This is the case in this cohort. The time elapsed between the last dose administration of anti CD20 and the first vaccination dose did not change this. We used different cut-off values (4, 5 and 6 months with similar results). We did not include these figure in the manuscript as we think most results are similar to others already published in the field, but would do if reviewer thinks it should be necessary.